# Treatment Planning Study for Microbeam Radiotherapy Using Clinical Patient Data

**DOI:** 10.3390/cancers14030685

**Published:** 2022-01-28

**Authors:** Kim Melanie Kraus, Johanna Winter, Yating Zhang, Mabroor Ahmed, Stephanie Elisabeth Combs, Jan Jakob Wilkens, Stefan Bartzsch

**Affiliations:** 1Department of Radiation Oncology, School of Medicine and Klinikum Rechts der Isar, Technical University of Munich (TUM), 81675 Munich, Germany; johanna.winter@helmholtz-muenchen.de (J.W.); yatingzhang@tju.edu.cn (Y.Z.); mabroor.ahmed@helmholtz-muenchen.de (M.A.); stephanie.combs@tum.de (S.E.C.); Jan.wilkens@tum.de (J.J.W.); stefan.bartzsch@helmholtz-muenchen.de (S.B.); 2Institute of Radiation Medicine (IRM), Helmholtz Zentrum München GmbH, German Research Center for Environmental Health, 85764 Neuherberg, Germany; 3Physics Department, Technical University of Munich (TUM), 85748 Garching, Germany; 4Partner Site Munich, Deutsches Konsortium für Translationale Krebsforschung (DKTK), 80336 Munich, Germany

**Keywords:** microbeam radiotherapy, spatial fractionation, treatment planning, dose calculation, equivalent uniform dose, software development

## Abstract

**Simple Summary:**

Microbeam radiotherapy is a novel dose delivery technique in radiation oncology. Preclinical studies have demonstrated preferable dose distributions with reduced damage to normal tissue but similar tumor control compared to conventional radiotherapy. For future clinical applications, realistic treatment plans for patient data are required as well as a method for comparing the spatially fractionated MRT doses with conventional broad beam doses. In this study, we performed MRT treatment planning on real patient data for relevant clinical scenarios. We successfully implemented a sophisticated dose comparison concept based on the equivalent uniform dose. For most scenarios and parameters studied, the clinical dose constraints were met. However, limitations were caused by the lack of treatment plan optimization and dose optimization. Altogether, we demonstrated the feasibility of achieving clinically acceptable MRT dose distributions based on real patient data as a primary major step towards clinical application of MRT.

**Abstract:**

Microbeam radiotherapy (MRT) is a novel, still preclinical dose delivery technique. MRT has shown reduced normal tissue effects at equal tumor control rates compared to conventional radiotherapy. Treatment planning studies are required to permit clinical application. The aim of this study was to establish a dose comparison between MRT and conventional radiotherapy and to identify suitable clinical scenarios for future applications of MRT. We simulated MRT treatment scenarios for clinical patient data using an inhouse developed planning algorithm based on a hybrid Monte Carlo dose calculation and implemented the concept of equivalent uniform dose (EUD) for MRT dose evaluation. The investigated clinical scenarios comprised fractionated radiotherapy of a glioblastoma resection cavity, a lung stereotactic body radiotherapy (SBRT), palliative bone metastasis irradiation, brain metastasis radiosurgery and hypofractionated breast cancer radiotherapy. Clinically acceptable treatment plans were achieved for most analyzed parameters. Lung SBRT seemed the most challenging treatment scenario. Major limitations comprised treatment plan optimization and dose calculation considering the tissue microstructure. This study presents an important step of the development towards clinical MRT. For clinical treatment scenarios using a sophisticated dose comparison concept based on EUD and EQD2, we demonstrated the capability of MRT to achieve clinically acceptable dose distributions.

## 1. Introduction

Microbeam radiation therapy (MRT) is a novel cancer treatment technique using spatially fractionated photon radiation that has first been studied for cancer therapy in the 1990s [1]. Several 10 micrometer-wide kilovoltage X-ray beams are spaced hundreds of micrometers apart, leading to a unique beam profile of high dose beamlets, called peak doses, and low doses in between, called valley doses. Peak doses of several hundred Gray can be delivered by third-generation synchrotrons providing sufficiently high photon fluxes and quasi-parallel beams [2]. Due to the dose–volume effect, a higher normal tissue dose tolerance can be expected for the micrometer-scaled dose distribution [3]. Preclinical data revealed improved normal tissue sparing using MRT, while target dose efficacy remained comparable to conventional broad beam radiation treatments (CRT) [4,5,6,7,8].

The biological effect of the spatially fractionated dose is not completely understood yet. In vitro studies demonstrated differences in the response of normal and tumor tissue towards MRT [9,10,11], while in vivo experiments showed high tumor control at reduced normal tissue toxicity [12,13,14]. Plausible mechanisms for a reduced normal tissue toxicity include a better coordinated repair of the more regular cellular architecture in normal tissue than in tumor tissue [15], a higher sensitivity of the tumor microvasculature towards MRT compared to CRT [16] and a different immune response after MRT than after CRT [10]. Translational studies on treatment planning, dose coverage and doses to organs at risks (OARs) are scarce and have mainly focused on phantom dosimetry [1,17,18,19,20,21]. Smyth et al. [22] simulated MRT dose distributions for clinical patient data and found a ratio of the peak dose to the valley dose (PVDR) above 10, which is essentially smaller than PVDRs from prior preclinical studies using smaller field sizes. Small or shallow tumors such as brain tumors, head and neck tumors and loco-regionally recurrent breast cancer sites were identified as potential future MRT targets. Larger field sizes result in lower PVDRs due to more scattered radiation and thus a higher valley dose. Deep-seated target volumes receive lower peak doses because of a steeper depth dose of kilovoltage X-rays compared to megavoltage X-rays, which also results in a lower PVDR [2]. Generally, a high peak dose is considered essential for tumor control in MRT, whereas a low valley dose ensures sparing of normal tissues. Smyth et al. therefore considered a PVDR of >10 as a minimum requirement for an MRT treatment. Since we are using the equivalent uniform dose (EUD), we do not need a minimum criterion for PVDR in this study. However, comparison of doses between CRT and MRT remains challenging since the translation of the spatially fractionated doses into clinical doses is not well understood. Most studies have focused on the valley dose as the parameter that correlates with normal tissue complication. However, in vivo data contradict the equivalence of valley doses with CRT doses [23], suggesting that the ratio of peak and valley doses and their spatial distribution have to be considered, too.

Multi-directional MRT can be implemented in different geometries. An interlaced geometry yields a rather homogeneous target dose, while the surrounding tissue receives a spatially fractionated dose [24]. However, interlaced MRT with a micrometer-precise alignment of the target volume is very challenging to implement in a clinical setting. In contrast, a cross-firing geometry has lower alignment demands and causes dose variations on the micrometer scale in the target volume and in OARs [25,26]. Reporting dose distributions on a micrometer scale is difficult to achieve, and a standard to interpret such dose distributions with respect to tumor control and toxicity has not yet been established. An interpretation of the dose distribution on a macroscopic computed tomography (CT) voxel grid by assigning a homogeneous dose equivalent to the microscopic dose pattern is desirable. Recently, it was suggested to use the EUD describing the dose leading to the same clonogenic cell survival according to the clinically well-established linear quadratic model [27]. The EUD concept has been reasonably successful in the description of moderately modulated dose distributions in CRT [28,29], and an evaluation for MRT and other forms of spatially fractionated radiation therapy is still pending.

The aim of this planning study was to identify suitable clinical scenarios for future application of MRT. We implemented the concept of EUD for MRT dose evaluation and simulated various MRT treatment scenarios for clinical patient data.

## 2. Materials and Methods

### 2.1. Patient Data

We chose five different patient cases that provided a variety of clinical applications in radiotherapy indications as well as tumor locations and sizes. Data were retrospectively acquired at the Department of Radiation Oncology of the university hospital of the Technical University of Munich. The tumors covered a glioma case, where in the clinical scenario the resection cavity was irradiated with 60 Gy in 2 Gy per fraction using eight different beam angles. Furthermore, we chose a case of radiosurgery for a shallowly located small sarcoma brain metastasis using 7 beam angles. These cases were chosen since brain tumors and metastases might be a suitable application for MRT as existing preclinical data have shown promising results. In addition, brain tumors can be fixated very accurately without the interference of organ motion, which will be a crucial aspect in future clinical application of MRT. We also included a small non-small cell lung cancer (NSCLC) case, which was treated with stereotactic body radiation therapy (SBRT) using eight beam directions. Recent studies of preclinical MRT lung tumor treatments showed promising results [13,30], and combinations with systemic drugs that are increasingly used for NSCLC treatment seem promising [31,32]. Furthermore, the rather small tumor volume and the peripheral location might be an advantage for MRT applications. A breast tumor treated with hypofractionated radiotherapy delivered by two tangential beams was chosen due to its shallow location that might be suited for MRT. Furthermore, as another clinically common scenario, a case of bone metastasis of the ribs using two opposed tangential beams was chosen. Dose regimes and tumor volumes are depicted in Table 1.

### 2.2. Microbeam Treatment Planning

For each patient, we simulated MRT dose distributions based on the CT and planning target volume (PTV) used for clinical treatment planning. For PTV definition, the same PTV margins were used for the clinical and MRT treatment plan and were delineated by experienced radiation oncologists. For the glioblastoma resection cavity, a margin of about 20 mm around the resection cavity was used. For the lung SBRT, an internal target volume (ITV) was defined based on the 4D-CT, and for PTV definition, an additional spherical margin of 10 mm was added. For palliative bone metastasis treatment, a margin of about 20 mm around the clinical target volume (CTV) was applied, and neighboring soft tissue was included according to the radiation oncologist’s assessment. For radiosurgery of the brain metastasis, a spherical margin of 1 mm was used, and for whole breast tumor treatment, we added 10 mm around the gross tumor volume (GTV). For all cases manual adaptation according to the radiation oncologist was performed if clinically required. Dose calculation was performed with hybridDC, a hybrid dose calculation engine combing the accuracy of Monte Carlo simulations for photon interactions and efficient kernel-based dose calculations for the electron transport [33]. Monte Carlo simulations of photon interactions yielded distributions of primary and scatter photon dose. The electron kernel algorithm converts both quantities to dose profiles based on the microbeam pattern. HybridDC is based on Monte Carlo simulations, and hence we report dose distributions as dose-to-medium.

We developed an MRT treatment planning tool as an add-on in the open-source platform 3DSlicer for a user-friendly application with a graphical user interface [34]. We applied built-in functions of 3DSlicer and of the open-source Radiotherapy module [35] to load CT datasets and dose distributions and also for the calculation of dose volume histograms and dose metrics. For faster dose calculation, we resized the CT voxels in *x*- and *y*-direction to double the original dimensions.

### 2.3. Microbeam Planning Parameters

We used parallel beams of synchrotron X-rays with the spectrum of the biomedical beamline ID17 at the European Synchrotron Radiation Facility in Grenoble, France, with a mean photon energy of 104 keV. The microbeam peak width was 50 µm, and the center-to-center distance was 400 µm. The number of beams and the gantry angles were adapted only if necessary in order to satisfy the tradeoff between target volume coverage and sparing of OARs. Currently, only coplanar beam arrangements can be considered with MRT. Multi-directional MRT was implemented in a cross-firing geometry of coplanar beams.

As multileaf collimators shape the radiation fields in clinical treatments, we also implemented conformal fields for the MRT dose calculation. The MRT beams were shaped as the respective target volume projected onto the beam direction. The projection of the target volume was dilated by one or two voxels for a full target volume coverage.

### 2.4. Dosimetric Evaluation and Equivalent Uniform Dose

For the comparison of micrometer-scaled MRT to clinical plans, we report the MRT dose distributions as *EUD* in the resolution of the CT voxels. For that purpose, we analyzed the microscopic dose distribution in hybridDC in a subvoxel resolution of 25 µm^3^. After sorting the subvoxel doses in increasing order, we arranged them into 35 equal groups and calculated the mean dose of each group to obtain a dose histogram with 35 bins for each CT voxel. From this histogram, we calculated the *EUD* in Python according to the linear quadratic model (LQM) [36] as
(1)$$EUD=-\frac{\alpha }{2\beta }+\sqrt{{\left(\frac{\alpha }{2\beta }\right)}^{2}-\frac{\ln\left(SF\right)}{\beta }}$$
with the survival fraction
(2)$$SF=\overset{n}{\underset{i=1}{\sum }}w_{i}e^{-\alpha D_{i}-\beta D_{i}{}^{2}}$$
where *α* and *β* denote the tissue-specific radiobiological parameters in the LQM; $w_{i}$. the volume fraction receiving dose *D_i_*, i.e., the histogram bin height, and n the total number of histogram bins. To each CT voxel, we assigned the respective *α*- and *β*-values based on the clinical contours from the Dicom dataset. The contours were extracted as labelmaps from 3DSlicer and read into Python. Most underlying values of *α* and *β* were extracted from reviews by Kehwar et al. [37] and van Leeuwen et al. [38], with additional values from other publications [3,39,40,41,42,43]. In cases of insufficient evidence, we chose a default *α*-value of 0.1 Gy^−1^ and a default *β*-value of 0.05 Gy^−2^ for normal tissue [44]. Note that in out-of-field regions with homogeneous scatter dose, the EUD corresponds to the sum of all scatter doses.

We assumed that MRT treatments are delivered in a single temporal fraction scenario. For comparison, all MRT EUD and clinical doses were converted into the equivalent dose in 2 Gy fractions (EQD2) based on the LQM according to
(3)$$\mathrm{EQD}2=D\frac{\left[d+\frac{\alpha }{\beta }\right]}{\left[2Gy+\frac{\alpha }{\beta }\right]}$$
where d denotes the dose per fraction and D the total dose of the clinical plan and the EUD of the MRT plan, respectively.

For each patient case, we compared the MRT EQD2_EUD_ with clinical EQD2_clinical_. We calculated dose volume histograms (DVHs) and extracted single dosimetric parameters as they were relevant for the investigated tumor. For all PTVs, we present the mean dose as well as the dose received by 2% and 98% of the PTV. For organs at risk, we compared results to dosimetric constraints from the literature [39,40,41,42,43,45] after conversion to EQD2.

## 3. Results

Dosimetric results revealed MRT as a comparable dose delivery method for the majority of clinical scenarios investigated in this work. Detailed dosimetric analyses for the PTVs and OARs are presented in Table 1 and Table 2, respectively.

Normalizing the MRT dose D_98%_ to D_98%_ of the clinical treatment in the PTV, for the case of the glioblastoma resection cavity irradiation, target coverage was achieved, almost all dose constraints for the relevant OARs were kept by MRT and the dose metrics were comparable to the clinical dose distribution, as depicted in Table 1 and Table 2 and Figure 1a. Only the maximum doses to the brain stem and the cochlea exceeded the constraints, which was caused by the limited flexibility to conform the dose to the PTV close to these OARs, as shown in Figure 2. The cochlea and the brain stem, which even overlapped the PTV, were spared by clinical treatment planning, whereas the MRT distribution was simulated conformal to the entire PTV. However, Figure 2 also shows the comparability of the two dose distributions for the conventional treatment plan (a,b) and the MRT plan (c,d).

For the lung SBRT, the MRT treatment plan achieved target coverage and satisfied most of the dosimetric constraints. However, for the trachea, the maximum dose was 47.32 Gy and the maximum dose to 0.1 cm^3^ was 19.00 Gy, as indicated in Table 2 and visualized in Figure 1b. The origin of this extreme maximum dose was a spatially very limited hotspot calculated in air within the trachea and can be explained by the low density of some voxels leading to few interaction events during Monte Carlo dose calculation and very limited actual energy absorption and yet high doses due to the low density. The maximum dose to the aorta exceeded 59 Gy and the maximum dose to the heart was up to 16.44 Gy with a mean heart dose of 1.86 Gy. The reasons for these high doses were broader, PTV-conformal fields in the MRT plan that overlapped behind the target volume, whereas in the clinical treatments, the dose was prescribed to the 60% isodose covering the PTV. Corresponding dose distributions are shown in Figure 3, where the differences in dose values can be seen. Higher entrance doses in the MRT plan (c, d) and a less conformal dose distribution are also visible.

For the bone metastasis, we found acceptable doses for OARs for both the clinical treatment plan and the MRT plan. All dose constraints were met. However, a general trend of increased doses for MRT was noted, as seen in Table 2 and Figure 1c. Compared to the clinical dose, the maximum dose to the myelon and the mean dose to the heart was 10 and 6 times higher than for the clinical plan, respectively. The maximum dose to 150 cm^3^ of the small bowel was also increased for the MRT plan. This can be partially explained by the dose normalization of the MRT PTV dose to fit the D_98%_ of the PTV for the clinical dose to achieve full target volume coverage and by the shallower PTV DVH curve for the MRT plan. Since the MRT PTV dose for the bone metastasis was less homogenous than for the clinical dose, the applied normalization resulted in increased doses to OARs.

When comparing a single fraction MRT dose delivery to a brain metastasis with radiosurgery, doses to OARs were higher for MRT, though far below critical dose limits. Only the maximum dose to the whole brain exceeded the dose constraint for the clinical as well as for the MRT treatment plan.

Comparing hypofractionated radiotherapy for breast cancer with MRT, we found acceptable doses for OARs for both treatment modalities. While the lung and the contralateral breast were better spared by the MRT treatment plan, the doses to the heart, liver and myelon were lower for the clinical treatment plan. However, the target dose was less homogenous for the MRT dose resulting in less steep DVHs, as shown in Figure 1e.

## 4. Discussion

We demonstrated that clinically acceptable treatment plans can be achieved with X-ray MRT in the kilovolt range. Moreover, we identified clinical scenarios suited for potential application of MRT. We established a novel method for dose comparison based on the EUD based on equivalent cell survival fractions in the linear quadratic model, an essential step to compare conventional and MRT doses. Furthermore, we implemented a basic treatment planning software for MRT that was integrated into the open-source toolkit 3DSlicer.

Our results showed the possibility to generate MRT plans that keep clinical dose constraints for OARs and achieve good target volume coverage. The feasibility of clinically acceptable treatment plans with orthovoltage X-rays is a milestone for the clinical development of MRT. Compared to the clinical treatment plans, there was a trend towards increased normal tissue doses. However, the higher OAR doses for the bone metastasis and for the breast cancer MRT plan arose mainly from limited options of beam shaping and weighting. For the dose distribution of the glioblastoma resection cavity, the increased doses to the cochlea and brain stem, which even overlapped the target volume, were caused by limited flexibility to conform the dose to the target volume and the missing dose optimization. Whereas the clinical dose is shaped by multi-leaf-collimators and optimized according to the dose prescription taking neighboring and overlapping OARs into account, the MRT doses in this study were simply shaped by the target volume for each CT slice. The absence of dose optimization allowing for intra-and inter-beam weighting is another major limitation that must be addressed in the future. However, considering these limitations, the results achieved within this study appear even more promising.

Clinical treatment planning systems use analytical dose calculations with electron-density scaling and provide the dose to water, whereas Monte Carlo-based algorithms calculate the dose to the specific medium, resulting in differing dose distributions in this study [46]. We chose not to convert the dose to water into dose to medium or vice versa because of additionally introduced uncertainties from the required stopping power ratios for each tissue type [47,48]. In addition, the dose error by omitting the conversion was reported below 4% for most tissues and only higher for cortical bone (up to 14%) [49]. A treatment planning study with head and neck and prostate cancer plans revealed deviations of the relevant dose metrics between 0% and 8%, also with highest deviations in hard bone [48]. For these reasons, our comparison between clinical and MRT doses seems valid as a first assessment of clinical MRT plans.

The complex microstructure of the lung might deteriorate the spatial dose fractionation on the same scale [50] but has not yet been considered in the MRT dose calculation algorithm. However, preclinical MRT studies provided promising results regarding tumor control and healthy tissue sparing [30]. The consideration of the lung microstructure in MRT treatment planning represents a challenging task since up-to-date CT data do not spatially resolve the microstructure. For this reason, the effect of the microstructure on the MRT dose distribution needs to be incorporated into the treatment planning algorithm. Furthermore, tumor and organ motion due to breathing and the heartbeat has not been considered within this study, but depending on the available dose rate, it may play an even more important role for MRT than for CRT due to smearing of the micrometer-scaled dose distribution [51].

Altogether, these results argue for clinical scenarios where correct dose calculation and conformation is less complex for first future application of MRT. Our results did not reveal obvious dosimetric arguments preferring one over the other selected cases as previously proposed by Smyth et al. [22]. However, the studies differ with respect to the evaluated beam and dose parameters. Whereas Smyth at al. only evaluated MRT dose distributions for a limited beam width and a fixed beam direction, we used clinically more realistic beam and treatment plan characteristics. Specification of PVDRs is only possible for simple beam setups but difficult for several crossing beams, where peak and valley doses are not clearly defined. While Smyth et al. analyzed valley doses for normal tissue tolerances, we used the derived EUDs that consider the entire spatially fractionated dose distribution and thus might represent the more precise model for dose comparison [27]. Clearly, the EUD calculation has limitations, too. Firstly, the applied LQM strongly depends on the tissue specific α- and β-values with variable evidence [38] and strong inter-study variability. Secondly, only repair mechanisms are taken into account by the LQM, and other biological effects of MRT such as immune response and bystander effects are not considered [52]. Thirdly, the LQM might not be appropriate for high fraction doses compared to the α/β-ratio [53]. However, there is evidence that the LQM is applicable for fraction doses up to 18 Gy [54,55]. Since in our study, most fraction doses were below 18 Gy, we consider the utilization of the LQM to be valid. The precise survival prediction in peak regions with extremely high doses is almost irrelevant for the overall cell survival and EUD estimate in the microbeam fields. Lastly, due to the lack of superior alternatives, the LQM might be the best concept for dose comparison of broad beam and spatially fractionated dose distributions to date.

For MRT in clinical routine, synchrotrons will be unsuitable due to their limited availability and missing clinical infrastructure, but compact MRT sources are needed. Microbeam treatment planning for such compact sources must consider their divergent X-ray field and possible additional hotspots from crossing of the divergent beams. The presented dose calculation algorithms are flexible and can be extended to such radiation fields.

## 5. Conclusions

For the first time, we simulated MRT dose distributions based on real patient data using clinical treatment plan parameters and compared the MRT with clinical dose distributions using the EUD concept. For most of the investigated cases, a glioblastoma resection cavity, a brain metastasis, a lung tumor, a bone metastasis and a breast tumor, we found clinically acceptable dose distributions for MRT. For the lung tumor, accurate consideration of the microstructure will be required. The presented EUD data need to be interpreted with caution and require a careful validation in preclinical studies. Improvement of the dosimetric results can be achieved by a more sophisticated treatment planning process including dose optimization.

## Figures and Tables

**Figure 1 cancers-14-00685-f001:**
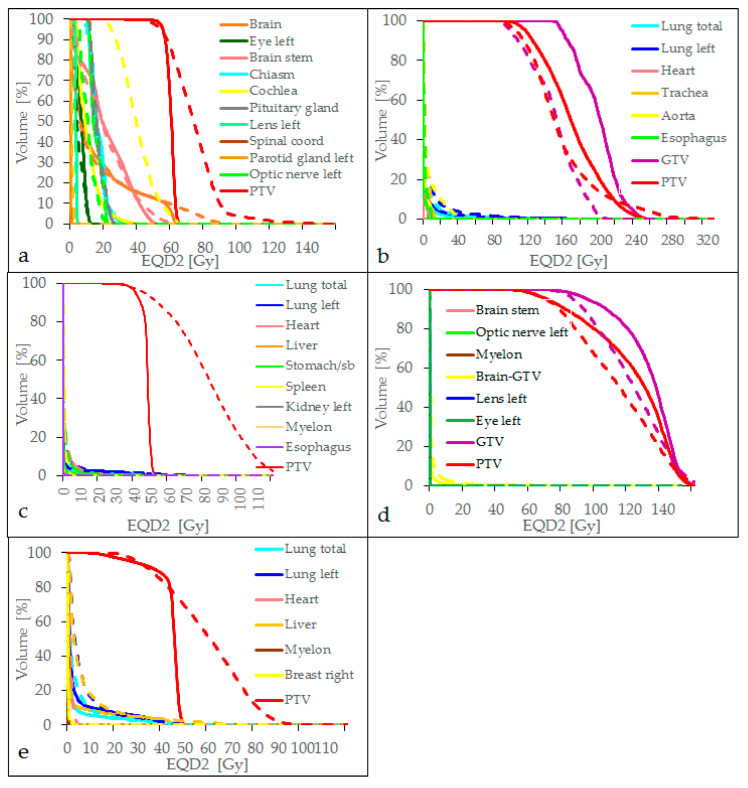
Dose volume histograms for the five clinical scenarios: (**a**) glioblastoma resection cavity, (**b**) lung SBRT, (**c**) sarcoma bone metastasis, (**d**) sarcoma brain metastasis, (**e**) breast cancer. Solid lines represent the clinical treatment plans, dashed lines the MRT plans.

**Figure 2 cancers-14-00685-f002:**
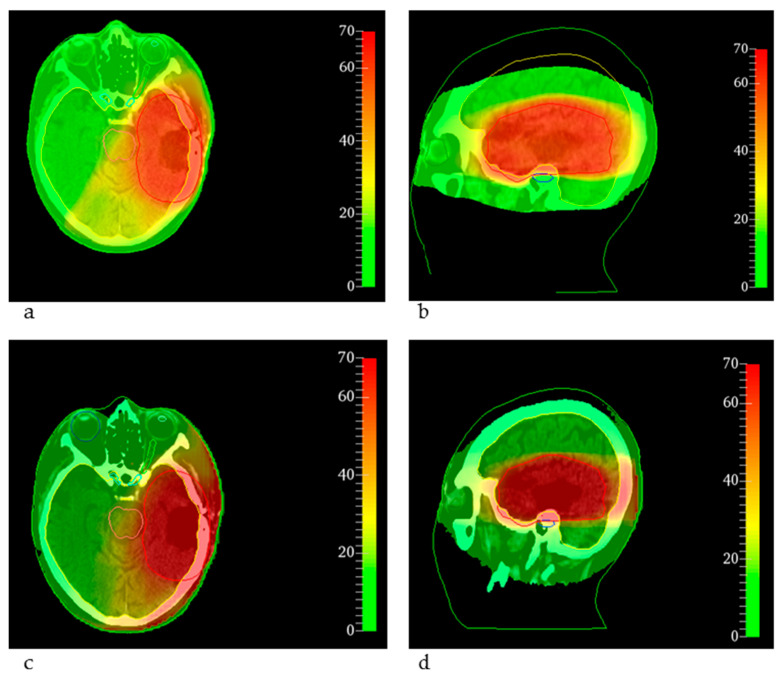
EQD2 distributions overlaid on the corresponding CT slice for the glioblastoma resection cavity. The color bar indicates the dose in Gy. (**a**,**b**) The conventional clinical dose distributions as EQD2 on a transversal and sagittal slice, respectively. (**c**,**d**) The corresponding EQD2 of the equivalent uniform dose for MRT. The overlap of the cochlea (blue) and the brain stem (light red) with the PTV (red) is shown. The brainstem is depicted in light red, the brain is depicted in yellow, the chiasm in cyan, the left optic nerve in green, the right eye in blue and the left eye in green.

**Figure 3 cancers-14-00685-f003:**
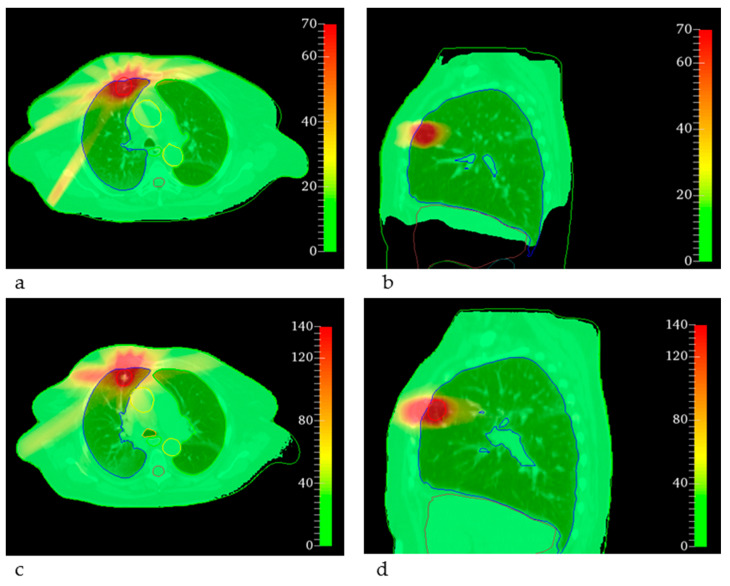
EQD2 distributions overlaid on the corresponding CT slice for the lung SBRT. The color bar indicates the dose in Gy. (**a**,**b**) The conventional clinical dose distributions as EQD2 on a transversal and sagittal slice, respectively. (**c**,**d**) The corresponding EQD2 of the equivalent uniform dose for MRT. The PTV is delineated in red, the right lung in blue, the aorta in yellow and the trachea in orange.

**Table 1 cancers-14-00685-t001:** Dose regimes, prescription doses D_prescription_ and the mean dose D_mean_ as well as doses to 2% and 98% of the PTV, D_2%_ and D_98%_ are reported. All dose values of the clinically applied fractionated doses and the corresponding calculated equivalent uniform doses for single fraction MRT are converted to fractionated dose calculated as 2 Gy equivalent doses (EQD2). SBRT stands for stereotactic body radiotherapy, RS stands for radiosurgery and GBM stands for glioblastoma.

Plan				Clinical D_prescription_	EQD2_clinical_				EQD_MRT_		
	α/β (Gy)	α (Gy^−1^)	β (Gy^−2^)		D_prescription_ (Gy)	D_98%_ (Gy)	D_mean_ (Gy)	D_2%_ (Gy)	D_98%_ (Gy)	D_mean_ (Gy)	D_2%_ (Gy)
GBM cavity	2.096	0.035	0.0167	60 Gy in 2 Gy fractions	60.00	53.05	59.57	63.82	50.11	74.98	115.29
Lung SBRT	10.0	0.3	0.03	37.5 Gy in 12.5 Gy fractions to 60% isodose	70.31	106.89	168.01	238.27	97.15	159.41	274.62
Sarcoma bone metastasis	3.00	0.0585	0.0195	39 Gy in 3 Gy fractions	46.80	38.54	47.61	51.24	38.77	82.22	120.50
Sarcoma brain metastasis RS	3.00	0.0585	0.0195	20 Gy in a single fraction	92.00	61.39	121.54	153.48	61.97	112.70	157.03
Breast tumor hypofractionated	4.20	0.1025	0.02631	40.05 Gy in 2.67 Gy fractions	44.38	18.04	43.71	48.87	25.46	59.41	89.93

**Table 2 cancers-14-00685-t002:** Dose constraints for organs at risk (OARs) for the fractionated dose calculated as 2 Gy equivalent doses (EQD2) and the corresponding calculated EQD2 values for the clinically applied treatment plan and for the equivalent uniform dose resulting from the simulated single fraction MRT dose delivery. GBM = glioblastoma, SBRT = stereotactic body radiotherapy, RS = radiosurgery, hf = hypofractionated, l = left, r = right. Dosimetric values that did not meet the required constraints are highlighted in bold letters.

Plan	OAR	α/β (Gy)	α (Gy^−1^)	β (Gy^−2^)	Dose Values (Gy) (%)		
					EQD2_constraints_	EQD2_clinical_	EQD2_MRT_
GBM cavity							
	Brain stem	2.096	0.035	0.0167	D_max_ < 54 Gy	D_max_ = 51.49 Gy D_mean_ = 21.39 Gy	**D_max_ = 65.79 Gy** D_mean_ = 16.58 Gy
	Cochlea	2.096	0.035	0.0167	D_max_ < 45 Gy	D_max_ = 39.97 Gy D_0_._1cm_^3^ = 16.58 Gy	**D_max_ = 65.92 Gy** D_0.1cm_^3^ = 20.95 Gy
	Chiasm	2.988	0.0251	0.0084	D_max_ < 55 Gy	D_max_ = 32.51 Gy	D_max_ = 30.08 Gy
	Optiv nerve r	2.994	0.0497	0.0166	D_max_ < 55 Gy	D_max_ = 10.40 Gy	D_max_ = 10.13 Gy
	Optic nerve l	2.994	0.0497	0.0166	D_max_ < 55 Gy	D_max_ = 22.67 Gy	D_max_ = 25.02 Gy
	Spinal cord	2.007	0.0307	0.0081	D_max_ < 50 Gy	D_max_ = 0.88 Gy	D_max_ = 2.35 Gy
	Pituitary gland	2.096	0.035	0.0167	D_max_ < 45 Gy	D_max_ = 26.96 Gy	D_max_ = 25.62 Gy
	Brain without PTV	2.096	0.035	0.0167	D_mean_ < 30 Gy	D_mean_ = 10.79 Gy	D_mean_ = 10.60 Gy
	Parotid gland l	2.991	0.0341	0.0114	D_mean_ < 26 Gy	D_mean_ = 0.62 Gy	D_mean_ = 1.29 Gy
	Lens l	1.002	0.0544	0.0543	D_max_ < 5 Gy	D_max_ = 4.68 Gy	D_max_ = 4.98 Gy
Lung SBRT							
	Heart	1.997	0.0579	0.029	D_max_ < 26 Gy	D_max_ = 1.25 Gy	D_max_ = 16.44 Gy D_0.1cm_^3^ = 14.94 Gy
	Trachea	2.00	0.1	0.05	D_max_ < 32 Gy	D_max_ = 7.92 Gy	**D_max_ = 47.32 Gy** D_0.1cm_^3^ = 19.00 Gy
	Aorta	2.00	0.1	0.05	D_max_ < 45 Gy	D_max_ = 23.21 Gy	**D_max_ = 59.11 Gy** D_0.1cm_^3^ = 54.10 Gy
	Esophagus	3.00	0.0585	0.0195	D_mean_ < 34 Gy	D_mean_ = 0.59 Gy	D_mean_ = 2.23 Gy
	Lung total	3.79	0.0307	0.0081	V_20Gy_ < 20%	V_20Gy_ = 2.38%	V_20Gy_ = 5.12%
	Lung ipsilateral	3.79	0.0307	0.0081	D_mean_ < 7 Gy	D_mean_ = 3.67 Gy	**D_mean_ = 7.70 Gy**
Sarcoma bone metastasis							
	Lung ipsilateral	3.79	0.0307	0.0081	D_mean_ < 7 Gy	D_mean_ = 1.20 Gy	D_mean_ = 2.36 Gy
	Myelon	2.007	0.0307	0.0153	D_max_ < 45 Gy	D_max_ = 0.03 Gy	D_max_ = 0.32 Gy
	Lung total	3.79	0.0307	0.0081	V_20Gy_ < 20%	V_20Gy_ = 1.04%	V_20Gy_ = 1.20%
	Heart	1.997	0.0579	0.029	D_mean_ < 26 Gy	D_mean_ = 0.29 Gy	D_mean_ = 1.73 Gy
	Stomach/ Small bowel	7.0	0.0895	0.0128	150 cm^3^ < 30 Gy	150 cm^3^ = 0.37 Gy	150 cm^3^ = 1.78 Gy
	Kidney ipsilateral	3.0	0.0106	0.0036	V_50%_ < 14 Gy	V_50%_ = 0 Gy	V_50%_ = 0 Gy
	Esophagus	3.00	0.0585	0.0195	D_mean_ < 30 Gy	D_mean_ = 0.03 Gy	D_mean_ = 0.20 Gy
Brain metastasis RS							
	Brain stem	2.096	0.035	0.0167	D_max_ < 54 Gy	D_max_ = 0.01 Gy	D_max_ = 0.57 Gy
	Optic nerve l	2.994	0.0497	0.0166	D_max_ < 55 Gy	D_max_ = 0.01 Gy	D_max_ = 0.38 Gy
	Myelon	2.007	0.0307	0.0153	D_max_ < 50 Gy	D_max_ = 0.00 Gy	D_max_ = 0.14 Gy
	Brain-GTV	2.096	0.035	0.0167	D_max_ < 60 Gy	**D_max_ = 155.24 Gy**	**D_max_ = 164.61 Gy**
	Lens l	1.002	0.0544	0.0543	D_max_ < 5 Gy	D_max_ = 0.00 Gy	D_max_ = 0.14 Gy
	Eye l	2.0	0.1	0.05	D_max_ < 45 Gy	D_max_ = 0.01 Gy	D_max_ = 0.22 Gy
Breast tumor hf							
	Lung total	3.79	0.0307	0.0081	V_20Gy_ < 20%	V_20Gy_ = 4.13%	V_20Gy_ = 3.70%
	Lung ipsilateral	3.79	0.0307	0.0081	D_mean_ < 7 Gy	D_mean_ = 4.12 Gy	D_mean_ = 6.35 Gy
	Heart	1.997	0.0579	0.029	D_mean_ < 4 Gy	D_mean_ = 0.25 Gy	D_mean_ = 1.28 Gy
	Liver	1.500	0.0683	0.0455	D_mean_ < 30 Gy	D_mean_ = 3.36 Gy	D_mean_ = 7.06 Gy
	Myelon	2.007	0.0307	0.0153	D_max_ < 40 Gy	D_max_ = 0.28 Gy	D_max_ = 1.06 Gy
	Breast contralateral	3.400	0.3	0.0882	D_max_ < 2.64 Gy	**D_max_ = 5.69 Gy**	**D_max_ = 3.64 Gy**

## Data Availability

Data is contained within the article.

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
