# Peer review of "Treatment Planning Study for Microbeam Radiotherapy Using Clinical Patient Data"

_cancers, 2022, doi:10.3390/cancers14030685_

Round 1
Reviewer 1 Report
- Please proofread and revise the submission abstract as there appears to be either a formatting or grammatical error for this section, particularly the first sentence.
- The authors discussed that small or shallow tumors are potential sites treatable with MRT, however, failed to explain to readers why these sites would be favorable targets with MRT. Please consider discussing in further depth the beam characteristics and physics behind why these superficial sites are favorable. This will enhance the understanding of this topic more by journal readers.
- For the selected cases, planning target volumes (PTV) were used as targets. A pertinent finding in this paper is that while most clinical dose constraints were met across all cases, due to limitations with ability to conform dose and optimization issues with MRT, several critical OARs were not able to have constraints met. There is no mention by the authors about how PTV margins were defined. A PTV margin for a palliative bone metastasis will be significantly larger than a lung SBRT or brain metastasis radiosurgery case. The authors should mention whether they are using either subsite-specific PTV margins and whether these are institutional preference or drawn from established protocols/literature. If a symmetric PTV margin was used across all cases, by using smaller PTV margins for more conformal treatment plans (lung SBRT and brain metastasis), this may reduce overlap with nearby OARs and make the authors dosimetric outcomes and subsequent findings even more salient.
Author Response
- Please proofread and revise the submission abstract as there appears to be either a formatting or grammatical error for this section, particularly the first sentence.
- We proofread the abstract and changed the first sentence.
- The authors discussed that small or shallow tumors are potential sites treatable with MRT, however, failed to explain to readers why these sites would be favorable targets with MRT. Please consider discussing in further depth the beam characteristics and physics behind why these superficial sites are favorable. This will enhance the understanding of this topic more by journal readers.
- We added the following explanation: “Larger field sizes result in lower PVDRs due to more scattered radiation and thus a higher valley dose. Deep-seated target volumes receive lower peak doses because of a steeper depth dose of kilovoltage x-rays compared to megavoltage x-rays, which also results in a lower PVDR.”
- For the selected cases, planning target volumes (PTV) were used as targets. A pertinent finding in this paper is that while most clinical dose constraints were met across all cases, due to limitations with ability to conform dose and optimization issues with MRT, several critical OARs were not able to have constraints met. There is no mention by the authors about how PTV margins were defined. A PTV margin for a palliative bone metastasis will be significantly larger than a lung SBRT or brain metastasis radiosurgery case. The authors should mention whether they are using either subsite-specific PTV margins and whether these are institutional preference or drawn from established protocols/literature. If a symmetric PTV margin was used across all cases, by using smaller PTV margins for more conformal treatment plans (lung SBRT and brain metastasis), this may reduce overlap with nearby OARs and make the authors dosimetric outcomes and subsequent findings even more salient.
- We used the clinically applied PTV contours for each treatment case and clarified this in the beginning of section 2.2 by adding the following paragraph:
“For PTV definition the same PTV margins were used for the clinical and MRT treatment plan and were delineated by experienced radiation oncologists. For the glioblastoma resection cavity, a margin of about 20 mm around the resection cavity was used. For the lung SBRT an internal target volume (ITV) was defined based on the 4D-CT and for PTV definition an additional spherical margin of 10 mm was added. For palliative bone metastasis treatment, a margin of about 20 mm around the Clinical target volume (CTV) was applied, neighbouring soft tissue was included according to the radiation oncologist’s assessment. For Radiosurgery of the brain metastasis, a spherical margin of 1 mm was used and for whole breast tumor treatment, we added 10 mm around the gross tumour volume (GTV). For all cases manual adaptation according to the radiation oncologist were performed if clinically required. “
For a more sophisticated MRT planning, individual PTV margins for each planning case might reduce doses to OAR but are beyond the scope of this study. We consider this point covered by the second paragraph in the discussion “limited flexibility to conform the dose to the target volume and the missing dose optimization”.
Reviewer 2 Report
Thank you for the interesting manuscript! The authors investigated the cytogenetic level, advanced radiotherapy techniques VMAT and IMRT with the conventional 3D-CRT, using biological dosimetry. The work is well written and deserves to be published after minor, but mandatory to my opinion, modifications.
I think the authors should test the significance between the plans in Table 1 and Table 2 to prove that the difference.
L212: please explain more detail about figure 2 and figure 3.
Author Response
Thank you for the interesting manuscript! The authors investigated the cytogenetic level, advanced radiotherapy techniques VMAT and IMRT with the conventional 3D-CRT, using biological dosimetry. The work is well written and deserves to be published after minor, but mandatory to my opinion, modifications.
- We investigated spatially fractionated radiotherapy, namely microbeam radiotherapy, and compared it to conventional (broad-beam) treatment plans (including IMRT, 3D-CRT, radiosurgery and SBRT). We did not use biological dosimetry (as for example presented by Treibel et al. Establishment of Microbeam Radiation Therapy at a Small-Animal Irradiator. Int J Radiat Oncol Biol Phys. 2021 Feb 1;109(2):626-636. doi: 10.1016/j.ijrobp.2020.09.039.) but rather radiobiological in vitro data as part of the equivalent uniform dose.
I think the authors should test the significance between the plans in Table 1 and Table 2 to prove that the difference.
- We consider that a significance test does not add a large value for our results in table 1 and table 2 since the number of parameters is way too small to achieve a valid significant result The purpose of this manuscript is not a (significant) difference between the microbeam treatment plan and the conventional treatment plan but rather a proof of comparability of MRT plans to established clinical plans.
L212: please explain more detail about figure 2 and figure 3.
- We added the following sentence for Figure 2 in section 3: “However, Figure 2 also shows the comparability of the two dose distributions for the conventional treatment plan (a,b) and the MRT plan (c,d).”
- We added the following sentence for Figure 3 in section 3: “Corresponding dose distributions are shown in Figure 3, where the differences in dose values can be seen. Also, higher entrance doses in the MRT plan (c, d) and a less conformal dose distribution is visible.”
Reviewer 3 Report
General comments:
This is a topical paper written by a group with demonstrated experience in microbeam radiotherapy. Clinical studies in this area are still lacking, thus dosimetric and radiobiological evaluations of treatment plans are critical before clinical implementation.
Clinically feasible treatment plans were simulated using a hybrid MC dose calculation on different lesions and fractionation regimens, and EUDs were determined for microbeam RT and compared with conventional RT.
The paper is generally well written and interesting. There are, however, some aspects that could be improved. Below are my comments / suggestions:
The Introduction should include a short paragraph on the plausible mechanisms behind normal tissue benefits from MRT, to justify the implementation of this therapy in clinical settings.
Define the parameter ‘peak-to-valley-dose ratio’ and explain the meaning of the PVDR > 10 reported by Smyth et al when compared to other studies.
Introduction - “Small or shallow tumours…. were identified as potential future MRT targets” – explain why.
Methods: as well-known from radiobiology, the alpha/beta ratio for normal tissue (late effects) is around 3Gy (and yours is 20Gy(!) according to the manuscript). In the paper you cite (Grün, R.; Friedrich, et al) the authors discuss RBE-weighted target dose distribution in carbon ion therapy, with no normal tissue effects presented. The authors should check the correctness of the reference used here. Also please explain the large discrepancy between the commonly used alpha/beta ratio and your choice of this parameter for normal tissue effects. Interestingly, table 2 reflects these values otherwise (i.e., the correct ones).
Specific comments:
- Simple summary – ‘limitations were caused by the lack of treatment plan…’ – what do you mean by this sentence?
- Abstract, line 27 – for which reduced normal tissue… (replace ‘that’ with ‘which’)
- Line 162- is it (25 um)3 or 25 um3 ?
- Line 281 – replace ‘majorly’ with ‘mainly’ or ‘mostly’
- Line 295 – we chose not to convert the…
Author Response
General comments:
The Introduction should include a short paragraph on the plausible mechanisms behind normal tissue benefits from MRT, to justify the implementation of this therapy in clinical settings.
- We added the following sentence: “Plausible mechanisms for a reduced normal tissue toxicity include a better coordinated repair of the more regular cellular architecture in normal tissue than in tumour tissue [Crosbie2010], a higher sensitivity of the tumour microvasculature towards MRT compared to CRT [Bouchet2013], and a different immune response after MRT than after CRT [Yang2014].”
Define the parameter ‘peak-to-valley-dose ratio’ and explain the meaning of the PVDR > 10 reported by Smyth et al when compared to other studies.
- We modified the following sentence in the introduction: „ Smyth et al. [22] simulated MRT dose distributions on clinical patient data and found the ratio of the peak dose to the valley dose (PVDR) peak-to-valley dose ratios (PVDRs) above ten,…”
- We added the following sentence in the introduction: ”Generally, a high peak dose is considered essential for tumour control in MRT, whereas a low valley dose ensures sparing of normal tissues. Smyth et al. therefore considered a PVDR of > 10 as minimum requirement for an MRT treatment. Since we are using EUD, we do not need a minimum criterion for PVDR in this study.”
Introduction - “Small or shallow tumours…. were identified as potential future MRT targets” – explain why.
- We added the following explanation: “Larger field sizes result in lower PVDRs due to more scattered radiation and thus a higher valley dose. Deep-seated target volumes receive lower peak doses because of a steeper depth dose of kilovoltage x-rays compared to megavoltage x-rays, which also results in a lower PVDR.”
Methods: as well-known from radiobiology, the alpha/beta ratio for normal tissue (late effects) is around 3Gy (and yours is 20Gy(!) according to the manuscript). In the paper you cite (Grün, R.; Friedrich, et al) the authors discuss RBE-weighted target dose distribution in carbon ion therapy, with no normal tissue effects presented. The authors should check the correctness of the reference used here. Also please explain the large discrepancy between the commonly used alpha/beta ratio and your choice of this parameter for normal tissue effects. Interestingly, table 2 reflects these values otherwise (i.e., the correct ones).
- We thank the reviewer for this important comment. The mentioned alpha and beta values were typos! We changed it to the used default values of alpha = 0.1 Gy-1 and beta =0.05 Gy-2 resulting in an alpha/beta ratio of 2 Gy. We also changed the literature reference to [Tommasino, F.; Scifoni, E.; Durante, M. New Ions for Therapy. International Journal of Particle Therapy 2016, 2, 428–438, doi:10.14338/IJPT-15-00027.1.].
Specific comments:
- Simple summary – ‘limitations were caused by the lack of treatment plan…’ – what do you mean by this sentence?
- For clarification, we added “optimization” after treatment plan.
- Abstract, line 27 – for which reduced normal tissue… (replace ‘that’ with ‘which’)
- We changed the first sentence since this was asked by another reviewer, too.
- Line 162- is it (25 µm)3 or 25 µm3?
- (25 µm)3 as stated in the text is correct. We used cubes with an edge length of 25 µm.
- Line 281 – replace ‘majorly’ with ‘mainly’ or ‘mostly’
- Done.
- Line 295 – we chose not to convert the…
- Done.